# An Investigation of Various Machine and Deep Learning Techniques Applied in Automatic Fear Level Detection and Acrophobia Virtual Therapy

**DOI:** 10.3390/s20020496

**Published:** 2020-01-15

**Authors:** Oana Bălan, Gabriela Moise, Alin Moldoveanu, Marius Leordeanu, Florica Moldoveanu

**Affiliations:** 1Faculty of Automatic Control and Computers, University POLITEHNICA of Bucharest, Bucharest 060042, Romania; alin.moldoveanu@cs.pub.ro (A.M.); marius.leordeanu@cs.pub.ro (M.L.); florica.moldoveanu@cs.pub.ro (F.M.); 2Department of Computer Science, Information Technology, Mathematics and Physics, Petroleum-Gas University of Ploiesti, Ploiesti 100680, Romania; gmoise@upg-ploiesti.ro

**Keywords:** fear classification, emotional assessment, feature selection, affective computing

## Abstract

In this paper, we investigate various machine learning classifiers used in our Virtual Reality (VR) system for treating acrophobia. The system automatically estimates fear level based on multimodal sensory data and a self-reported emotion assessment. There are two modalities of expressing fear ratings: the 2-choice scale, where 0 represents relaxation and 1 stands for fear; and the 4-choice scale, with the following correspondence: 0—relaxation, 1—low fear, 2—medium fear and 3—high fear. A set of features was extracted from the sensory signals using various metrics that quantify brain (electroencephalogram—EEG) and physiological linear and non-linear dynamics (Heart Rate—HR and Galvanic Skin Response—GSR). The novelty consists in the automatic adaptation of exposure scenario according to the subject’s affective state. We acquired data from acrophobic subjects who had undergone an in vivo pre-therapy exposure session, followed by a Virtual Reality therapy and an in vivo evaluation procedure. Various machine and deep learning classifiers were implemented and tested, with and without feature selection, in both a user-dependent and user-independent fashion. The results showed a very high cross-validation accuracy on the training set and good test accuracies, ranging from 42.5% to 89.5%. The most important features of fear level classification were GSR, HR and the values of the EEG in the beta frequency range. For determining the next exposure scenario, a dominant role was played by the target fear level, a parameter computed by taking into account the patient’s estimated fear level.

## 1. Introduction

According to statistics, 13% of the world’s population is affected by phobias, a type of anxiety disorder manifested by an extreme and irrational fear towards an object or a situation. 275 million people suffer from anxiety disorders throughout the world and anxiety disorders are ranked as the 6th-most common contributors to global disability [1]. Phobias are classified into social phobias (fear of relating to others or speaking in public) and specific phobias (generated by particular objects or situations). Social phobias affect people of all ages, though they usually start to manifest in adolescence. 17% of people with social phobias develop depression. The majority of them turn to medication, and even substance abuse and illegal drugs (nearly 17%) or alcohol (nearly 19%), and only 23% seek specialized help [2]. With regard to specific phobias, a significant percent (15–20%) of the world’s population faces one specific phobia during their lifetime [3]. The most common specific phobias and their prevalence are: acrophobia (fear of height)—7.5%; arachnophobia (fear of spiders)—3.5%; aerophobia (fear of flying)—2.6%; astraphobia (fear of lightning and thunder)—2.1%; and dentophobia (fear of dentist)—2.1%. [4]. Specific phobias begin during childhood and can persist throughout one’s life, affecting more women than men. Most of these patients do not seek treatment for phobias and, of those who do, only 20% recover completely [2].

The treatment for phobias is either medical or psychological. 80% of people suffering from phobias turn to medicines and Cognitive Behavior Therapy (CBT), a form of psychotherapy that encourages patients to modify destructive patterns of cognition and behavior and to replace them with positive thoughts [5]. Immersion therapy consists of gradual exposure to anxiety-producing stimuli, in the presence of the therapist who controls the intensity of immersion [6]. Thus, the patients are urged to understand their fears and find a way to adjust their attitude towards the anxiety-provoking object/situation in a conscious and apperceptive fashion. The medical or psychological treatment should be continued for as long as required since statistics reveal that phobia tends to relapse in approximately 50% of cases [7]. With the technological advancement, Virtual Reality has significantly emerged in recent decades, allowing the design of immersive virtual worlds that provide stimuli in a safe and controlled manner [8].

In 1997, Picard published a seminal book entitled Affective Computing, in which are presented the theories and principles of a new interdisciplinary field encompassing computer science, neuroscience, psychology, and engineering [9]. Affective Computing (AC) is defined as “computing that relates to, arises from, or influences emotions”.

According to Picard, computers need to understand human emotions and even have and express emotions for the purpose of communicating with humans. AC enables an integration of human emotions into technology. The field comprises: the study of affect recognition and generation methods, expressing affection techniques, affect aware systems development, research on the modality in which affect influences human-technology interactions. AC helps people understand psychological phenomena, human behaviors, and to build better software applications [10]. AC has many applications in education, game development, health, robots, cyber-psychology, VR, marketing, entertainment, and so on. 

The integration of affective information in game development opens the path to new methods of maintaining players’ engagement [11], by dynamically adjusting game levels difficulty to tailor the users’ individual emotional characteristics [12]. In healthcare applications, AC involves automatic emotion detection and provides decisions accordingly. Relational agents have been developed in order to help patients in hospitals or to assist childbirth, offering information and emotional support [13]. Conversational agents and robots interact with children suffering from ASD, helping them to develop from the socio-emotional point of view [14]. 

In this paper, we propose a VR game for treating acrophobia, based on the idea of real-time automatic adaptation of in-game height exposure according to the subject’s level of fear. With physiological signals as input (EEG, GSR and HR), our system determines the subject’s current fear level and predicts the next exposure scenario. 

The current fear level and the next exposure scenario were obtained using various machine learning (ML) and deep learning (DL) classifiers: Support Vector Machine (SVM), k-Nearest Neighbors (kNN), Linear Discriminant Analysis (LDA), Random Forest (RF), and 4 deep neural network architectural models. The data used for training the classifiers was recorded in a preliminary experiment in which 8 acrophobic subjects were in vivo and virtually exposed to various heights. For computing the accuracy of the classifiers, both a user-dependent and a user-independent modality were used. Therefore, each classifier was trained using the data of the other subjects in the case of the user-independent modality. We calculated cross-validation and test accuracies applying the trained model on the data of the tested user. Moreover, this is the research idea towards which we are inclined, given the fact that training a classifier for every subject is an unfeasible and highly time-consuming activity. On the other hand, in the case of the user-dependent modality, for each subject, each classifier was trained using their own data, obtained from the preliminary experiment. The trained model was then applied on the test records of the same participant. Feature selection was also computed for each classifier in order to improve generalization across subjects.

To validate our method, we performed an experiment with the same 4 acrophobic users, in which they played the proposed acrophobia game twice. The first classifier predicted the current fear level, while the second one estimated the next exposure scenario (or game level to be played next). The results showed a very high cross-validation accuracy on the training set (obtained by the kNN and RF classifiers) and good test accuracies, ranging from 42.5% (for the 4-choice scale) to 89.5% (for the 2-choice scale) (both for SVM, for the player-dependent modality). Also, we determined that the most relevant features for fear level classification were GSR, HR and the values of the EEG in the beta frequency range. For the next exposure scenario prediction, an important role was played by the target fear level.

The paper is organized as follows: Section 2 presents the state of the art regarding the current VR-based therapies, Section 3 introduces a short description of the emotional models and types of physiological signals employed in our research, Section 4 details similar experiments and the modalities in which various machine learning techniques have been used for emotion classification, Section 5 presents our acrophobia game, together with the ML and DL approach for fear level and next exposure scenario prediction, Section 6 provides an insight into the methodology for training, dataset construction and experimental procedure, while Section 7 emphasizes the results of our experiments. Finally, we discuss the research findings in Section 8 and present the conclusions and future work directions in Section 9.

## 2. VR-Based Phobia Therapy

Virtual Reality has been involved in phobia treatment since the 1990s. In the study presented in [15], 60 participants suffering from agoraphobia have been equally divided into two groups: a control group and an experimental group. Eight virtual environment scenes were used to expose 30 participants from the experimental group in session sequences of about 15 min. The Attitude Towards Agoraphobia Questionnaire (ATAQ) and Subjective Unit of Discomfort (SUD) were used as instruments to assess the anxiety states of the subjects. SUD means decreased over the eight sessions, from 5.66 to 2.42, indicating habituation with the agoraphobic stimuli. The results proved that agoraphobic patients can be successfully treated with VR technologies. VR technologies have manifold applications in phobia treatment, from understanding the causes of these disorders, to evaluating and treating them [16,17]. 

Virtual Reality Exposure Therapy (VRET) is a behavioral treatment for anxiety disorders, including phobias. The patient is immersed in a computer-generated virtual environment which presents stimuli that are dangerous in real-world situations [18]. VRET is equally as efficient as the classical evidence-based interventions (CBT and in vivo exposure), provides real-life impact, has good stability of results in time and engages the patients in the therapy as much as in vivo exposure does [8,16]. The existing VRET systems can be classified into platforms, academic research projects or experiments and mobile/desktop game applications.

### 2.1. Platforms

C2Phobia [19] was designed by mental health professionals, psychiatrists, psychologists and psychotherapists. Using a VR headset, the patient is gradually exposed to anxiogenic situations. The system can also be used at home, allowing the specialist to treat patients and prescribe personal exercises at a distance. C2Phobia is recognized as a medical device, a complete therapeutic software, but the developers did not disclose whether they use machine learning techniques or not. 

PSIOUS [20] provides animated and live VR and Augmented Reality (AR) environments, as well as 360-degree videos for anxiety disorders, fears and phobias treatment. It offers patients monitoring capabilities, generation of automatic reports and the possibility of home training. PSIOUS contains 70 VR scenes. The developers did not disclose whether they used machine learning techniques or not.

Stim Response Virtual Reality [21] provides a wide range of virtual worlds for acrophobia, aerophobia and social phobias treatment, as well as physiological data synchronization. The VR and AR scenes change in real time, based on the subject’s responses to the environment. It performs automatic data analysis.

Virtual Reality Medical Center [22] uses 3D virtual environments, biofeedback and CBT to treat phobias and anxieties (especially pre-surgical anxiety), relieve stress, and teach relaxation skills. Non-invasive physiological monitoring with visual feedback allows control for both the patient and the therapist. Virtually Better [23] is a system available only for the therapist’s office which is aimed at providing therapeutic applications for treating phobias, job interview anxiety, combat-related post-traumatic stress disorders, and drug or alcohol addiction. Virtually Better has been used by the VR Treatment Program at Duke Faculty [24], where the therapist guided the participants through the environment and interacted with them through the entire event. Research studies have indicated that 6 to 12 45–50-minute-long therapy sessions were enough to achieve maximum benefit. The Bravemind system [25] was used for treating soldiers who served in Iraq and Afghanistan with anxiety disorders. It works by providing vibrotactile and olfactive sensations associated with war zones. Limbix [26] contains interactive scenes made of panoramic images and videos that can be changed by the therapist in real-time. Lastly, PHOBOS [27] was designed in consideration of CBT protocols. It provides interactive environments, gradual exposure and realistic crowd and social group dynamics simulations for treating social and specific phobias. 

### 2.2. Academic Research Projects and Experiments

Acrophobia Therapy with Virtual Reality (AcTiVity-System—UniTyLab, Hochschule Heilbronn, Germany) [28] is played on an Android device and uses the Oculus Rift headset [29] to render 3D scenes, Microsoft Kinect [30] for motion tracking, and a heart rate sensor for measuring HR. The virtual environment contains buildings that have a walk route on the sides. A large experiment was performed in which 100 users were divided into a VR group and a control group. The participants from the VR group had to take a tour in a 10-storey office complex. All 44 subjects from the VR group who completed the six sessions of the experiment had an average reduction of 68% of their fear of heights. VR Phobias [31] presents a static virtual environment depicting the view from the balcony of a hotel. The results of an experiment in which 15 acrophobic patients were exposed to heights in vivo and virtually showed that the success rates of both procedures were similar. However, the VR exposure sessions were shorter, safer and more comfortable for the patients. The acrophobia system presented in [32] contained three virtual environments in a cityscape. The results of an experiment in which twenty-nine subjects participated and rated their fear levels in the presence of a therapist who adjusted exposure according to their affective state showed that both anxiety and avoidance levels decreased. Virtual therapy proved to be as effective as in vivo exposure to fear-provoking stimuli.

### 2.3. Mobile/Desktop Game Applications

Some of the most popular desktop game applications available for the Oculus Rift [29] and HTC Vive [33] headsets are The Climb [34], Ritchie’s Plank Experience [35], Arachnophobia [36] and Limelight [37]. The first two try to overcome fear of heights, Arachnophobia treats fear of spiders, while Limelight puts the user in front of a crowd with changeable moods where he/she gives lectures or presentations, in order to overcome their fear of public speaking. Samsung Fearless Cityscapes [38] and Samsung Fearless Landscapes [39] are dedicated to acrophobia therapy and are rendered via Gear VR [40] glasses. Hear rate can also be monitored when they are paired with Gear S2 [41].

Most of the VR applications mentioned above do not provide any details related to the technologies and the methods used. Thus, we cannot ascertain whether ML techniques were deployed for adapting the therapy. On the other hand, we are interested in building Machine Learning-based applications tailoring therapy to the individual characteristics of each patient. 

Our system detects the fear level in real time and automatically selects the next exposure scenario. By training the classifiers in a user-independent way with the data obtained from in vivo and virtual experiments, we aim to construct robust classification models that would accurately evaluate the patients’ affective states and adjust the levels of exposure accordingly. Thus, we intend to provide a reliable therapeutic solution for phobia alleviation based on Virtual Reality and human-centered machine learning. Our system can be used in clinics, for home therapy and deployed on mobile devices, incorporating all the advantages of the above-mentioned systems. 

## 3. Emotion Models and Physiological Data

### 3.1. Emotion Models

Emotion is defined as a feeling deriving from one’s circumstances, mood or relationship with others [42]. It is a complex psycho-physiological experience generated by the conscious or unconscious perception of an object or situation [43], manifested through bodily sensations and changes in mood and behavior. The bodily sensations originate from the autonomic nervous system (increased cardiac activity, dilatation of blood vessels, involuntary changes in the breathing rate), cortex (activation of emotion-related brain areas) and are accompanied by physical expressions such as tremor, crying or running [44]. Various classifications of emotions have been proposed, from both the discrete and the dimensional perspective. One of the first categorizations identified six discrete emotions: happiness, sadness, fear, anger, disgust and surprise [45]. Consequently, the list was updated with embarrassment, excitement, contempt, shame, pride, satisfaction, amusement, guilt, relief, wonder, ecstasy, and sensory pleasure [46]. Complex emotions can be constructed from a combination of basic emotions. Plutchik introduced the wheel of emotions to illustrate how basic emotions (joy versus sadness; anger versus fear; trust versus disgust; surprise versus anticipation) can be mixed to obtain different emotions [47]. Plutchik’s model is not the only tool used to assess emotional reactions. The Geneva emotional wheel (GEW) uses a circular structure with the axes defined by valence (bipolar subjective evaluation of positive/negative) and control to arrange 40 emotion terms in 20 emotion families [48]. The dimensional models organize emotions within a space with two or three dimensions along which the responses vary. Russell’s Circumplex Model of Affect [49] encompasses valence on the x-axis, indicating the positive or negative component of emotion and arousal along the y-axis, reflecting the degree of mental activation or alertness that is elicited [50]. Arousal ranges from inactive (not excited) to active (excited, alert) [43]. Besides valence and arousal, a third dimension, called dominance, specifies the degree of control the subject exerts over the stimulus. Dominance ranges from a weak, helplessness feeling to a strong, empowered one. For instance, fear is defined as having low valence, high arousal and low dominance. From the behavioral decision-making perspective, we mention the approach-withdrawal (or appetitive-aversive) motivational model which reflects the tendency of approaching or rejecting the stimulus. According to [51], fear is generated by an aversive response that conducts to either active or passive physical reactions.

### 3.2. Physiological Data

Electroencephalography (EEG) non-invasively measures electrical potentials produced by neural activity which falls in the frequency ranges corresponding to the delta (<3 Hz), theta (3 Hz–8 Hz), alpha (8 Hz–12 Hz), beta (12 Hz–30 Hz) and gamma (>30 Hz) waves. EEG offers high precision time measurements—it can detect brain activity at a resolution of one millisecond—but unfortunately lacks spatial resolution. The recording area of an electrode is approximately one centimeter of the scalp, which corresponds to hundreds of thousands of neurons in the cerebral cortex. Thus, it is difficult to accurately pinpoint the exact source of brain activity or to distinguish between activities occurring at contiguous locations [52]. Moreover, EEG signals are prone to electrical interferences or artefacts resulting from body movements (eye blinks, muscular or cardiac activity) or environmental causes. 

The right hemisphere processes negative emotions or aversive behaviors, while the left one is involved in mediating positive emotions or approach behaviors. The people who experience negative feelings, who are angry, afraid or depressed, present activations in the amygdala (part of the limbic forebrain) and in the right prefrontal cortex [53]. The literature largely supports the approach/withdrawal model of alpha asymmetry, which states that activation in the right cortical area (low alpha waves) is associated with an aversive behavior, while activation in the left cortical area indicates positive feelings [54,55,56,57]. 

Park et al. [58] observed an increase of the beta waves at the left temporal lobe when the users experienced fear. The work of [59] showed reduced beta power in the bilateral temporal and right frontal cortex for individuals suffering from panic disorders. An increase of beta intensity in the left temporal lobe was also noticed in [58] whenever the subjects felt threatened.

The research performed by [60] showed that the patients who experienced fear exhibited high theta, delta and alpha absolute power and low beta levels. The authors suggest that the increase of the alpha waves accompanies and regulates the excessive excitation of the slow waves in the temporal regions and in the limbic system. In [61], a patient suffering from agoraphobia and panic attacks had an increase in the beta activity and a sudden decrease of frontal-central theta power. Time-domain EEG analysis indicated a reduced P300 Event Related Potential (ERP) and an increase in the beta activity in the right temporal lobe, an increase in the alpha activity in F4 and a decrease of the T5 theta activity [62]. 

In [63], a negative relationship was observed between delta and alpha 2 activity. A decreased beta-delta coherence in anxious individuals was shown in [61], together with a significant decrease in delta during panic attacks. Beta activity in the central part of the frontal cortex increased, being accompanied by a significant reduction of the theta waves all over the cortex, similar to what has been found in [64].

The ratio of slow waves to fast waves (SW/FW) has a negative correlation with fear [65,66,67]. There was a statistically significant reduction in the SW/FW ratio (delta/beta and theta/beta) in the left frontal lobe in an experiment where data has been recorded from a single electrode [68]. Neutral states are reflected in equal levels of activation in both hemispheres [69]. Quantitative EEG studies, and in particular coherence (linear synchronization between EEG signals measured at different brain locations), indicated a lower degree of inter-hemispheric functional connectivity at the frontal region and intra-hemispheric at the temporal region [70].

Plethysmography (PPG) is a non-invasive circulatory assessment method that uses an infrared photoelectric sensor to record changes in blood flow from the finger or from the ear lobe. It determines blood volume pulse by calculating how much of the emitted light is reflected back. The PPG values are converted into heart rate, which is measured in beats per minute (bpm). Heart rate variance is a strong indicator of emotion. In [71], a decrease in variance while the heart rate was high was an indicator of fear. Heart rate, combined with other variables, can successfully classify emotions [72,73], although in others it was found that it had the smallest contributing factor [74]. 

Electrodermal Activity (EDA) or Galvanic Skin Response (GSR) is a measure of sweat glands production and therefore skin activity, in direct relation with the sympathetic nerve’s state of excitation. GSR has two main components: tonic skin conductance, the baseline value recorded when no emotional stimulus is applied and phasic skin conductance, the response acquired when environmental and behavioral changes occur [50]. Increased GSR indicates arousal. It was the main contributing factor for emotion classification in several studies, including [75,76], being effective for discriminating fear from other negative emotions [77]. GSR recording devices are comfortable for users due to their light, easily attachable sensors [78].

In conclusion, we consider that the most relevant physiological signals to account for in fear assessment experiments are GSR, HR and the values of the alpha, beta and theta waves. In addition, the ratio of slow/fast waves is a good indicator of fear, together with alpha asymmetry—the difference in cortical activation between the right and left hemisphere in the alpha frequency band.

## 4. Physiological Data in VR-Based Machine Learning Applications for Treating Phobias

Virtual Reality can induce the same level of anxiety as real-life situations, and physiological data can be used to reflect stress level. In this section, we perform a short review on physiological data analysis in VR and on the ML techniques involved in emotion recognition and phobias treatment.

### 4.1. Physiological Data in VR-Based Applications for Treating Phobias

In the study presented in [79], the authors investigated the physiological responses of both nonphobic and phobic subjects in the VR environment. They monitored the skin resistance (SR), heart rate (HR) and skin temperature of 36 participants suffering from fear of flying and 22 participants with no fear. The anxiety level of the phobic participants was evaluated using Subjective Units of Distress, on a scale from 0 to 100 (0—no anxiety, 100—highest anxiety). The results showed a significant difference in the case of SR between two groups and no major difference in the case of HR and skin temperature. More intensive VR-based therapy sessions applied on the phobic subjects had a greater effect on 33 persons who succeeded to fly by plane after the VR treatment. 

More physiological data was recorded in the experiment performed in [80], which confirmed the following hypotheses: virtual heights increased the subjects’ stress levels and the cognitive load during beam-walking was higher in VR. Heart rate variability, heart rate frequency power, heart rate, electrodermal activity and EEG data have been recorded and analyzed to validate the two hypotheses. Heart rate variability varied from 6.6 beats/min in the unaltered low view to 7 beats/min in low VR conditions and 8.3 beats/min in high VR conditions. Heart rate started from 92 beats/min in unaltered view, continued with 97 beats/min in VR low and 97.1 beats/min in VR high conditions.

Electrodermal activity of five subjects was analyzed in [81] to measure stress level in VR conditions. The participants have not been diagnosed with acrophobia, but they claimed a certain discomfort in height situations. Each subject underwent a 15 min session consisting of three sub-sessions: height exposure in the real world (standing on the balcony of a building); height exposure in VR (the users did not interact with the VR environment); and height exposure in VR with VR environment interaction. The results proved that interaction with the environment during phobia treatment is important and that physiological measurements help in assessing emotional states.

Human responses to fear of heights in immersive VR (IVR) conditions were investigated in [82]. The authors performed two experiments: the first experiment on 21 subjects with ages ranging from 20 to 32 years and the second on 13 subjects with ages in the interval 20–27 years. During the first experiment, in which the subjects were exposed to four heights: 2, 6, 10, and 14 m in IVR conditions, GSR, heart rate and the participants’ view direction were measured. In the second experiment, the subjects were exposed till 40 m in an immersive virtual environment. The authors measured physiological responses and head motion. Also, the participants had to report the perceived anxiety level. The results showed that there was a correlation between the anxiety level and the subjects’ head pitch angle and that the anxiety level is accurately visible in phasic skin conductance responses. Also, it was established a correlation between anxiety/height and GSR measurements. 

### 4.2. Machine Learning for Emotion Recognition

Automatic emotion recognition has gained the attention of many researchers in the past few decades. As of now, there are three major approaches to automatic emotion recognition: the first approach consists in analyzing facial expressions and speech, the second approach uses the peripheral physiological signals, and the third approach uses the brain signals recorded from the central nervous system [83]. Certainly, a method that will embrace all these three approaches will provide the best results. The emotion recognition models are used in applications such as man-machine interfaces, brain-machine communications, computer-assisted learning, health, art, entertainment, telepresence, telemedicine and driving safety control [84,85,86].

Machine Learning offers computers the ability to learn from large data sets [87]. Among the ML techniques, Deep Learning is increasingly used in various applications, due to its higher accuracy when huge amounts of data are used for training. For emotion recognition, different ML techniques have been employed. 

A research tool called the Multimodal Affective User Interface is proposed in [85] for emotion discrimination. To obtain an accurate and reliable recognition tool, the system’s inputs were “physiological components (facial expressions, vocal intonation, skin temperature, galvanic skin response and heart rate) and subjective components (written or spoken language)” [85]. Using short films as stimuli for eliciting emotions and the GSR, temperature and heart rate records from 29 subjects, the authors implemented three ML algorithms: kNN, Discriminant Function Analysis (DFA) and Marquardt Backpropagation (MBP), in order to obtain six classes of emotions (sadness, anger, surprise, fear, frustration and amusement). The reported recognition accuracies were: kNN—67% for sadness, 67% for anger, 67% for surprise, 87% for fear, 72% for frustration and 70% for amusement; DFA—78% for sadness, 72% for anger, 71% for surprise, 83% for fear, 68% for frustration and 74% for amusement; MBP—92% for sadness, 88% for anger, 70% for surprise, 87% for fear, 82% for frustration and 83% for amusement. Also, the authors pointed out “that detection of emotional cues from physiological data must also be gathered in a natural environment rather than in one where emotions are artificially extracted from other naturally co-occurring states” [85].

A stack of three autoencoders with two softmax classifiers was used in the EEG-based emotion recognition system proposed in [86]. 230 power spectral features of EEG signals extracted in 5 frequency bands (theta, lower alpha, upper alpha, beta and gamma) and the differences between the spectral powers of all the 14 symmetrical pairs of electrodes on the right and on the left hemispheres have been used as inputs for some DL networks. The efficiency of the system was evaluated in four experimental setups: DLN-100 using a DL network with 100 hidden nodes on each layer; DLN-50 using a DL network with 50 hidden nodes; DLN-50 with PCA (Principal Component Analysis to address the overfitting problem); and DLN-50 with PCA and CSA (Covariate Shift Adaptation to solve the problem of non-stationarity in EEG signals). The accuracies obtained for each experiment were: DLN-100: 49.52% for valence and 46.03% for arousal; DLN-50: 47.87% for valence and 45.50% for arousal; DLN-50 with PCA: 50.88% for valence and 48.64% for arousal; DLN-50 with PCA and CSA: 53.42% for valence and 52.03% for arousal.

A comprehensive review of physiological signals-based emotion recognition techniques is presented in [88]. 16 studies including various classifiers such as Support Vector Machine, Linear Discriminant Analysis, k-Nearest Neighbors, Regression Tree, Bayesian Networks, Hidden Markov Model, Random Forest, Neural Networks, Canonical Correlation Analysis, Hybrid Linear Discriminant Analysis, Marquardt Back Propagation, Tabu search, and Fisher Linear Discriminant Analysis are compared with respect to their accuracies, bio-signal data, stimuli employed and feature extraction techniques. Emotions are considered in two models: discrete and dimensional. In the case of user dependent systems, the best performance (accuracy 95%) was achieved using linear discriminate in a novel scheme of emotion-specific multilevel dichotomous classification (EMDC) for joy, anger, sad and pleasure classification [89]. The bio-signals used were: and Electromyogram, Electrocardiogram, Skin Conductance, Respiration. An accuracy of 86% was obtained to classify joy and sadness in the case of user independent system [90]. The ECG feature extraction was performed using a non-linear transformation of the first derivative and tabu search was involved to acquire the best combination of the ECG features. 

Bayesian classifiers are used in a multimodal framework for analysis and emotion recognition [91]. Eight emotional states: anger, despair, interest, pleasure, sadness, irritation, joy and pride were recognized based on facial expressions, gestures and speech. The authors reported that all emotions except despair can be recognized with more than 70% accuracy and the highest accuracy was recorded for anger recognition (90%) [91].

A Deep Convolutional Neural Network-based approach for expression classification on the EmotiW (The Emotion Recognition in the Wild contest) dataset is presented in [92]. Seven basic expressions (neutral, happy, surprised, fearful, angry, sad and disgusted) were recognized, with an overall accuracy of 48.5% in the validation set and 55.6% in the test set.

The usage of VR environments as stimuli for human emotion recognition has barely been studied. In most research regarding automatic recognition of human emotions, the stimuli were either images, sequences of films or music. One of the first reports on EEG-based human emotion detection using VR stimuli is presented in [93]. Four deep neural networks were tested: standard, deep network with dropout, deep network with L1 regularization and deep network with dropout and L1 regularization. The last one achieved a 79.76% accuracy. Also, a high classification accuracy, close to 96%, was obtained for excitement detection while being immersed in a VR environment.

In [11], the physiological data of 20 Tetris players were recorded and analyzed using three classifiers: LDA, Quadratic Discriminant Analysis (QDA) and SVM. The results showed that playing the Tetris game at different levels of difficulty gives rise to different emotional states. Without feature selection, the best classifiers obtained an accuracy of 55% for peripheral signals and 48% for EEG (LDA, followed by SVM). Feature selection increased the classification accuracy to 59%, respectively, 56%. After the fusion of the two signal categories, the accuracy increased to 63%. A comparative study of four popular ML techniques aimed at identifying the affective states (anxiety, engagement, boredom, frustration and anger) of users solving anagrams or playing Pong is presented in [94]. The authors reported that SVM with a classification accuracy of 85.81% performed the best, closely followed by RT (83.5%), kNN (75.16%) and Bayesian Network (74.03%) [94]. A Dynamic Difficulty Adjustment (DDA) of game levels based on physiological data is presented in [12]. The authors used psychological responses during gameplay and a RT-based model for recognizing anxiety levels (low, medium, high). The model gave 78% correct predictions [12]. However, the adjustment was based on clauses and conditions, not on a prediction method.

A more detailed investigation of ML techniques used in emotions classification was performed in [95]. 

Fourteen physiological signals were recorded in VR conditions and used for emotion recognition in [96]: EEG f4, vertical and horizontal Electrooculography (EOG), Electromyography (EMG), Electrodermal Activity (EDA), Electrocardiogram (ECG), Chest Respiration (RIP), Abdomen Respiration (RIP), Peripheral Temperature, Heart Rate via PulseOx, Blood Volume (PPG) via PulseOx, Blood Oxygen (SpO2) via PulseOx, Head Acceleration and Rotation, Body Acceleration and Rotation. The Naive Bayes, k-Nearest Neighbor and Support Vector Machine techniques have been used to perform a binary classification: high-arousal or moderate/low arousal. The best accuracy was achieved in the case of SVM (89.19%).

### 4.3. Machine Learning for Identifying Anxiety Level in Phobia Therapy

In [97], a deep convolutional network was used to detect acrophobia level (level 1 = only somewhat strong or not strong, level 2 = moderately strong, level 3 = quite strong, level 4 = very strong). However, a tailored treatment was not performed. Richie’s Plank Experience was used as the virtual environment, and EEG data from 60 subjects was acquired to feed a deep learning network model VGG-16. The performance of the model has been measured using the accuracy, recall and precision parameters. The average accuracy obtained was 88.77%. 

A VRET system used to overcome public speaking anxiety, fear of heights and panic disorder is described in [98]. The system contains a mental stress prediction framework, which uses data extracted from GSR, blood volume pressure (BVP) and skin temperature signals to predict anxiety level. 30 persons participated in the experiments from [98], focused on public speaking anxiety. Four classes were defined for anxiety level: low, mild, moderate and high, and a SVM classifier with radial basis function (RBF) as kernel was used to train the models with various window lengths: 3, 5, 8, 10, 13, 15, 18, 20, 23, 25, 28, 29, and 30 s. A comparison between models was performed, and the results highlighted that the model using signal fusion outperformed the models using standalone signals. The early fusion method achieved the best accuracy of 86.3%. Model training and data processing were not performed during the experiments (Table 1).

Currently, VRET is seen as an efficient method for phobia treatment, both from a financial and a comfort point of view. It offers flexibility, confidentiality and trust, encouraging more people to seek treatment [16,96]. 

As far as we know, the issue of classifying emotion levels in VR conditions, meaning how intensely an emotion is felt based on different factors, has not been yet properly defined. 

In the proposed system, we focus our study on the ML and DL methods, which automatically classify fear level using physiological data. The dataset has been acquired in direct relation to our acrophobia therapy application, more specifically, by exposing the users to different heights in both the real-world and virtual environment.

## 5. The Machine Learning and Deep Neural Networks Approach for the Acrophobia VRET Game

The proposed VR system contains an ML-based decision support that adjusts the playing scenario according to the patient’s level of fear. It incorporates a real-time decision engine which uses the patient’s physiological data and determines the game level to be played next. In our ML-based decision support, the data obtained from the users contribute to configuring the game in order to suit each patient’s individual characteristics.

For this purpose, two classifiers were used: one to estimate the patient’s current fear level (C1) and one that determines the appropriate treatment according to the target fear level (C2). In our previous approaches [99,100], we used only deep neural networks as classifiers, but the obtained results pushed us to continue to test with various ML techniques. In this paper, we extended our work by defining a ML-based decision support that relies on various ML techniques such as SVM, kNN, LDA, RF and 4 deep neural network models (Figure 1). 

As in our previous work, we used two different fear level scales [99,100]:-2-choice scale, with 2 possible values, 0 and 1. 0 stands for relaxation and 1 stands for fear.-4-choice scale, with 4 possible values (0–3). 0—complete relaxation, 1—low fear, 2—moderate fear and 3—high level of anxiety.

The game scenarios consist of different game levels, each game level corresponding to a certain degree of height exposure in different contexts or a combination of certain height exposure degrees. The data recorded in real time from the patient is fed to the C1 classifier and the current fear level (FL_cr_) is computed. C1 estimates the level of fear the patient currently experiences.

To determine the target fear level (FL_t_) that ensures a gradual and appropriate exposure to height, we used the following formulas:
**2-choice scale****4-choice scale**if FLcr = = 0 then FLt = 1if FLcr = =1 then FLt = 0   if FLcr = = 0 or FLcr = = 1 then FLt = FLcr + 1   if FLcr = = 2 then FLt = FLcr   if FLcr= = 3 then FLt = FLcr − 1

The target fear level (FL_t_), together with the patient’s physiological data, are fed to the C2 classifier and the next game scenario (GS_pr_) is predicted. C2 estimates the phobia treatment.

The user plays the predicted level of the game and new physiological data is acquired. C1 computes a new general fear level and C2 predicts the game scenario to be played next. The process goes on for as long as the patient or the therapist consider appropriate—the patient can exit the game at any time if he/she feels uncomfortable—or a total predefined number of scenarios is reached. 

## 6. Experimental Methodology

The experiment was conducted in summer–autumn 2018 and involved the participation of 8 subjects who played an acrophobia game while their physiological (HR and GSR) and EEG data were recorded. The experiment was approved by the ethics committee of the UEFISCDI project 1/2018 and UPB CRC Research Grant 2017 and University POLITEHNICA of Bucharest, Faculty of Automatic Control and Computers. Prior to the experiment, the subjects signed a consent form and filled in a demographic and a Visual Height Intolerance questionnaire [101]. Prior to the tests, they were informed about the purpose of the experiment and research objectives. Moreover, they were presented with the steps of the procedure and the experimenter made sure that they fully understood what they were required to do. From the 8 users (aged 22–50 years, 6 women and 2 men), 2 suffered from a mild form of acrophobia, 4 from a medium-intensity fear of heights and 2 experienced a severe form of height intolerance. This classification resulted by assessing the responses to the Visual Height Intolerance questionnaire. More details can be found in [99,100]. They did not consume coffee or other energizing drinks before the experiment and made sure they had a relaxing sleep in the previous night. With respect to the therapy history, our subjects have not undergone any phobia alleviation treatment beforehand, neither medical nor psychological. Half of the users had previous experience in using VR systems and the others had not. For the second category, we provided some VR introductory sessions to accommodate them with the VR perception. Thus, we explained to them what a VR environment represents, which are the hardware components (VR glasses, controllers, sensors), how they work and how they can be adjusted. We presented the users the actions occurring in the game when each of the buttons from the controllers are pressed. Then, the subjects played a basic demo game which accommodated them with the VR perception.

The EEG data have been acquired using the Acticap Xpress Bundle [102] device with 16 dry electrodes, while HR and GSR have been recorded via Shimmers Multi-Sensory [103]. The next exposure scenario has been predicted in real-time by C2, based on the EEG, physiological data and the target fear level. The target fear level was calculated according to the formulas mentioned above, by taking into account the patient’s current fear level. The current fear level was estimated by C1. 

The classifiers we used were: kNN, SVM with linear kernel, RF, LDA and 4 deep neural network models with a varying number of hidden layers and neurons per layer.

### 6.1. Experiments and Dataset Construction

The 16 dry electrodes of the Acticap Xpress Bundle device [102] were placed according to 10/20 system in the following locations: FP1, FP2, FC5, FC1, FC2, FC6, T7, C3, C4, T8, P3, P1, P2, P4, O1 and O2. The log-normalized powers of all the 16 channels in the alpha, beta and theta frequency ranges were recorded and pre-processed in real-time for artefact removal. The ground and reference electrodes were attached to the ears. Using the Shimmer3 GSR+ Unit [104] of the Shimmers Multi-Sensory device, we acquired the subjects’ electrodermal activity and heart rate values. The Shimmer3 GSR+ Unit, which has Bluetooth connectivity, measures the skin’s electrical characteristics in microSiemens and captures a PPG signal (using the Shimmer optical pulse probe) that is later converted to estimate heart rate (HR). 

The two classifiers C1 and C2 have been fed with training data originating from two preliminary experiments where the subjects have been both in vivo and virtual y exposed to the first, fourth and sixth floors of a building, as well as on the ground floor, at 4 m, 2 m and a few centimeters away from a terrace’s railing. In the virtual environment, the players have been also exposed to the view from the building’s rooftop. The experiment in the virtual environment (consisting of three sessions, expanded over three days) has been preceded and succeeded by a real-world session. The EEG, GSR and HR data has been collected, together with the user’s perceived level of fear, called Subjective Unit of Distress (SUD), during each trial. Each patient was required to rate his/her fear on the 11-choice scale, a gradual scale with values from 0 to 10, where 0 corresponds to complete relaxation and 10 to extreme fear. The modality of reporting the SUD was verbally for the in vivo experiment and by pointing a virtual laser with the controller on a panel in the virtual environment (Figure 2).

The acrophobia game was rendered on the HTC Vive head-mounted display [33]. Interaction in the virtual environment was ensured through the controllers, so that the player advances in the game by teleportation—he/she presses on the floor in the virtual environment at various positions where he/she wants to go, is free to navigate wherever he/she wants, but he/she has to accomplishes the tasks of collecting coins of different colors (bronze, silver and gold) at 4 m, 2 m and 0 m distance from the balcony’s railing at ground level and at the first, fourth, sixth floors, as well as on the roof of the building. A coin is collected by bending and grabbing it with the controller. The game contained only visual and vestibular stimuli. There were no audio cues or animations to accompany the graphical presentation.

In both the real-world and virtual environment, each user totalized several 63 trials (3 sessions × 5 building levels × 3 distances from the railing = 45 in the virtual environment and 2 sessions × 3 building levels × 3 distances from the railing = 18 in the real-world environment). We thus obtained a dataset of 25,000 entries on average for each patient, which was saved in a database and used for training classifiers C1 and C2. 

For training C1, we had as input features the physiological data recorded during the 63 trials—the EEG log-normalized powers of the 16 channels in the alpha, beta and theta frequencies, the GSR and HR values. The output feature was the fear level (SUD) on three scales: the 11-choice scale (values as they were recorded, ranked from 0 to 10), 4-choice (fear rates from 0 to 3) and 2-choice (values of either 0 or 1). 

The ratings from the 11-choice-scale have been grouped into 4 clusters in order to create the 4-choice-scale (Table 2):-0 (relaxation)—rating 0 in the 11-choice-scale-1 (low fear)—ratings 1–3 in the 11-choice-scale-2 (medium fear)—ratings 4–7 in the 11-choice scale-3 (high fear)—ratings 8–10 in the 11-choice scale

Similarly, the ratings from the 4-choice-scale have been grouped into 2 clusters in order to create the 2-choice scale:-0 (relaxation)—ratings 0–1 in the 4-choice scale-1 (fear)—ratings 2–3 in the 4-choice scale.

*The classifiers we used were*: kNN, SVM with linear kernel, RF, LDA and 4 deep neural network models with different numbers of hidden layers and neurons per hidden layer. We have chosen these classifiers because they have been widely used in the literature (see Section 4.2 and Section 4.3). SVM provides the best results for emotion classification. kNN is used for signal classification. LDA has been used for binary and multi-class classification, being highly employed in the medical field. RF is a top classifier and the deep neural networks provide good classification results due to their ability to learn high-level features from large amounts of data in an incremental way.

The Sequential Forward Selection (SFS) feature selection algorithm was applied for kNN, RF and LDA. kNN is a non-parametric, feature similarity-based method used especially for classifying signals and images. The decision is made by taking into account the class of the majority of the k-nearest neighbors. SVM is a supervised machine learning algorithm that finds the hyperplane best segregating two or more classes. RF operates by constructing an ensemble of decision trees. The predicted class is obtained by combining the prediction of all individual trees, based on the “bagging” method stating that a combination of learning models increases the overall result. LDA is a dimensionality reduction technique that projects the dataset onto a lower dimensional space and finds the axes that maximize the separation between multiple classes, avoiding overfitting and reducing the computational cost. All these algorithms were run in Python, using their corresponding implementations from the scikit-learn library [105].

Using the TensorFlow deep learning framework [106], we created four Keras [107] sequential models for binary and multi-class classification: DNN_Model_1, DNN_Model_2, DNN_Model_3 and DNN_Model_4 (Table 3). Each network has an input layer of 50 neurons (16 neurons for the alpha values, 16 for the beta values, 16 for the theta values, 1 for GSR and 1 for HR) and an output of one neuron, corresponding to the predicted level of fear. Before training, the data has been standardized to reduce it to zero mean and unit variance. We performed a 10-fold cross-validation procedure 10 times and saved the weights of each network in .hdf5 files, together with the corresponding accuracies. The 10-fold cross-validation procedure was computed using the functionalities implemented in the scikit-learn library for k-fold cross-validation, with k = 10. The procedure has one parameter k that represents the number of groups the data is split into. Each group is taken as a test data set and the remaining k − 1 groups are taken as training data set. Then, the model is fit on the training set and tested on the test set. The evaluation score is retained, and the model is discarded. In the end, the cross-validation accuracy is calculated based on the k evaluation scores computed at each step.

Finally, the model version with the highest accuracy for each network has been selected and further used in the experiment. This procedure was repeated for every user, for the 2-choice, 4-choice and 11-choice scales. This technique was applied and published in [99,100]. In the current stage of research, we also trained ML classifiers (kNN, RF, LDA and SVM) in the same way—for every user, 10 times, for each fear scale—and the model providing the highest accuracy was saved for further use.

Classifier C2 predicts the game level that should be played next, i.e., the next exposure scenario (parameter GS_pr_). The Sequential Forward Selection (SFS) feature selection algorithm has been applied for kNN, RF and LDA. For training C2, the deep learning and machine learning models received as inputs the EEG, GSR, HR and SUD values, while the output represented an encoding of the height where these physiological values have been recorded—0 for ground floor, 1 for the first floor, 2 for the fourth floor, 3 for the sixth floor, and 4 for the roof of the building. For testing classifier C2, we provide as input EEG, GSR, HR and target fear level (FLt) and obtain as output the encoding of the height where the player should be taken to in the game (from 0 to 4, as mentioned above). Thus, if the user is currently feeling anxious (FLcr = 3), we calculate a target fear level FLt = 2 (so we want him to feel less anxious) and feed this value as input to classifier C2 in order to generate for us the next exposure scenario GSpr, on a scale from 0 to 4: 0 for ground floor, 1 for the first floor, 2 for the fourth floor, 3 for the sixth floor and 4 for the roof of the building.

The same DNN models were used for classifier C1, with the same number of hidden layers and neurons on each hidden layer (Table 3). Each network had an input layer of 51 neurons (16 neurons for the alpha values, 16 for the beta values, 16 for the theta values, 1 for GSR, 1 for HR and 1 for the “target fear level” feature). The output represented the level in the building from where the user should restart playing the game. The method for obtaining a personalized height exposure model to be validated on test dataset was: we repeated the 10-fold cross-validation procedure 10 times for each subject and saved the weights and the corresponding accuracies of each network in .hdf5 files; the model version with the highest accuracy for each network has been selected and further used in the experiment for all fear scales. 

The ML classifiers (kNN, RF, LDA and SVM) were trained in the same way—for every user, 10 times, for each fear scale—and the model resulting in the highest accuracy was saved for further use. For cross-validation, the data has been divided into 70% training and 30% test.

For computing the accuracy of the classifiers, both a user-dependent and a user-independent modality were used. Each classifier was trained using the data of the other subjects in the user-independent modality. We applied the trained model on the data of the tested user in order to calculate cross-validation and test accuracies. This approach makes it possible to calculate the performance of the classifiers in the worst possible case, where the model lacks user specificity. On the other hand, in the case of the user-dependent modality, for each subject, each classifier has been trained, cross-validated and tested on his/her own data. Feature selection has been also computed for each classifier in order to improve generalization across subjects. We used Sequential Forward Selection (SFS), a greedy algorithm that reduces the d-dimensional space to a k-dimensional space. In our case, we set k to 20, so that it would extract the most relevant 20 features from the total number of 50 features (16 EEG channels for the alpha, beta and theta waves, GSR and HR). The goal of feature selection was two-fold: we wanted to improve the computational efficiency and to reduce the generalization error of the model by removing irrelevant features or noise. SFS has been applied for kNN, RF and LDA.

### 6.2. The Acrophobia Game

The the game, which has been developed using the Unity engine [108], was synchronized in real-time with the Open Vibe [109] application for collecting EEG signals and with the Shimmer3 GSR+ Unit that records GSR and HR via Lab Stream Layer (LSL) [110]. Using a multi-thread C# application, we ran 5 threads simultaneously: one for recording the input from the game (fear ratings, events from the game, such as when the coins have been collected or when a level has been finished), peripheral physiological data (HR and GSR from the Shimmers3 Unit), alpha, beta and theta power spectral densities. At each session, 5 separate log .csv files were generated, each of them containing the timestamps (a timestamps represents the number of milliseconds passed since 1st January 1970) and the recorded data (either from the game, peripheral, alpha, beta or theta). The EEG data is extracted at an interval of 62.5 ms and the GSR and HR values were extracted at an interval of 19.53 ms. As the data has been saved at different sampling frequencies, we developed another processing module that merged the information from the log .csv files, averaged and aligned them according to the timestamps in order to have a compact dataset of EEG and peripheral recordings mapped onto the events occurring in the game. 

In order to extract the EEG data, we applied a bandpass Butterworth temporal filter, time-based epoching with the epoch duration of 1 s, and then squared the input values using the Simple DSP box from Open Vibe Designer. In addition, we averaged the signal and applied log-normalization using again the Simple DSP box. After all this preprocessing of the raw data, the alpha, beta and theta frequency powers have been extracted (Figure 3). 

All data was denoised and preprocessed in real time by applying a method named “the last correct value”, introduced by us. In our preprocessing module, all EEG and physiological data were inspected in real-time. As LSL was pulling sample data from the recording devices, before saving it into the corresponding log files, it was inspected to see if faulty values occur. For instance, if a negative value or one exceeding one and a half than the average of the previous values on a 5-seconds time span appeared, it was replaced with the average of the data recorded in the previous 5 s. If the device malfunctioned since the moment it started recording (suppose it took a longer time to initialize or calibrate), we initialized the last correct value with some average values—4.5 microVolts^2^ for the EEG log-normalized powers, 1 microSiemens for GSR and 75 bpm for HR. This method has been applied because we could not manually remove the noisy data nor stop the recording whenever such type of artefacts occurred in real-time.

In addition, it was saved in a database in both processed and unprocessed format for ulterior study and analysis. At the start of the game, the user was placed on the ground floor, where he/she had to navigate freely in the scene and collect a bronze, a silver and a gold coin (Figure 4 and Figure 5). The Shimmers Unit has been attached to the left hand and the right hand has been used for holding the HTC Vive controller. In this way, we tried to reduce the chances of introducing hand movement artefacts in the GSR and HR signals. At all time, the users were required to sit on a chair and move only their head and the right hand. 

Consequently, he/she reported the perceived SUD by pointing with a virtual laser on a panel which contained a range of options from 0 to 10 for fear level evaluation. The physiological data were averaged and classifier C1 predicted the subject’s current level of fear. To validate the accuracy of C1, we collected self-estimated SUDs. C1 predicted the current fear level based on the classification model created using the data from the previous experiment and a measure of assessing its accuracy was by comparing its output with the SUD perceived and acknowledged by the users directly during gameplay (during each trial of the game, we also asked the users to report the perceived fear level (SUD)). This parameter was called test accuracy. Based on the EEG, physiological data and target fear level (obtained using the fear level estimated by classifier C1), the next level of exposure was determined by classifier C2, either on the 2-choice or on the 4-choice scale. 

## 7. Results

The subjects played the game twice—once using the 2-choice and once using the 4-choice model. During each session, they were exposed to and played 10 scenes. At any time, the users could interrupt the game if they felt uncomfortable or the experimenter could terminate the session whenever he/she observed any abnormal events occurring. However, this was not the case, as all the subjects succeeded in completing both sessions without any difficulties. The maximum cross-validation accuracies on the training dataset and the validation (test) accuracies for each model, for both the player-independent and player-dependent modalities, with and without feature selection, are presented in Table 4, Table 5, Table 6 and Table 7. The Test column for the C2 classifier is empty because we did not use any method for testing the accuracy of C2. This classifier has only been cross-validated on the training dataset.

With SFS feature selection, the most selected features were: for the 2-choice scale—alpha FC2, C3, T8, O2, beta P4, theta C3, T8 and HR; for the 4-choice scale—alpha FC5, C3, T8, P4 and O2, beta FP2, FC5, P4 and theta T8; for the 11-choice scale—alpha FC2, C3, T8, beta FP2, C5 and HR. We observed that the most important features where the alpha values in the right pre-frontal area, left central and right temporal, beta values in the frontal and parietal areas, theta values in the temporal area and the heart rate.

The RF algorithm adds the benefit of computing the relative importance of each feature on the prediction. The implementation in the scikit-learn library measures feature importance by looking at how much the tree nodes using that feature reduce impurity for all the trees in the forest. Table 8 presents the most relevant 15 features, in descending order according to their importance, for the 2-choice, 4-choice and 11-choice scales, for both classifiers, for the player-independent modality. Table 9 contains the same attributes, but for the player-dependent modality. FL_t_ stands for “target fear level”, B_ for “beta”, A_ for “alpha” and T_ for “theta”. Thus, B_C3 represents the beta value of the C3 electrode (central scalp position, left side). A_FC6 represents the alpha value of the FC6 electrode (fronto-central position, right side).

For each relevant feature, we counted the total number of times it appeared across the RF classification model for the 2-choice, 4-choice and 11-choice scales. The maximum is 3 for a feature that is relevant for training on all the three fear estimation scales. 

The most relevant features for all 3 fear level estimation scales, for the user-independent modality, for Classifier 1 were: B_T8, A_FP1, T_FC6, B_FP1, B_FC5, A_FC6, B_FC6, B_FC2, B_P3, B_C3, HR and GSR. For Classifier C2, the most relevant features were: B_FP1, B_O2, B_FC2, B_C3, B_T8, B_P3, T_FC6, B_FC5, A_FC6, B_FC6, HR, FL_t_ and GSR. With respect to the user-dependent modality, for Classifier C1, for all 3 fear estimation scales, the most relevant features were: B_P3, B_C3, B_O1, T_FC6, B_FP1, A_FC6, B_P2, A_FP1, B_FC6, B_C4, B_FC2, HR and GSR. With respect to classifier C2, we mention: B_O2, B_FC2, T_FC6, B_FP1, B_FC6, A_FC6, GR, FL_t_ and GSR.

## 8. Discussion

The results presented in Table 4, Table 5, Table 6 and Table 7 show that the cross-validation and test accuracies obtained after SFS feature selection are lower than those obtained without feature selection. In Table 10, we present the classifiers providing the highest cross-validation and test accuracies for both the player-independent and player-dependent cross-validation and testing methods, on the 2-choice, 4-choice and 11-choice fear scales.

With respect to C1, the classifier predicting fear level, we conclude that the highest cross-validation accuracy (over 98%) was obtained by using either the kNN or RF algorithms, for both the player-independent and player-dependent modalities. The same trend occurs for C2, the classifier predicting the game level to be played next, where very high cross-validation accuracies were recorded by the RF classifier. With respect to the test (or validation) accuracy, for the 2-choice scale, the highest accuracy was obtained by DNN_Model_4 (79.12%) for the player-independent modality and SVM (89.5%) for the player-dependent modality. In the case of the 4-choice scale, the highest accuracies were provided by kNN (52.75%) and SVM (42.5%), respectively. We observed that SVM was very efficient for the player-dependent modality. For the 2-choice scale, both accuracies (79.12% and 89.5%) were higher than the random value of 50% when selecting either 0 or 1 by chance. The same happens in the case of the 4-choice scale, where the random, “by chance” accuracy is 25%. Both the kNN and SVM classifiers provided an accuracy higher than 25% (kNN—52.75% for the player-independent modality—and SVM—42.5% for the player-dependent method).

Both the player-independent and player-dependent training and testing modalities offered good classification results, making it difficult to determine which was best. However, we incline towards using the player-independent one, as we want a more general, less user-specific model.

With respect to features importance (determined by the RF classifier), we observed that GSR, HR and the beta waves play a significant role in fear level prediction for C1. They are followed closely by the alpha and theta activations, but on a lower extent. In the case of C2, the classifier predicting the game level to be played next, the “target fear level” feature, the feature we computed based on the user’s current fear level plays a dominant role, not only because it has a high feature importance index determined by the RF classifier, but also because it is selected when using all three fear level estimation scales (2-choice, 4-choice and 11-choice), for both the player-independent and player-dependent modalities. Our findings are in line with the state-of-the-art literature supporting the idea that GSR, HR and the beta waves are related to emotions classification, particularly fear assessment [111,112]. As there are no experiments in which the next game level is predicted based on physiological data, we cannot compare the results obtained by cross-validating and testing C2. However, it is worth pointing out that the same GSR, HR and EEG features are elicited and, in addition, to emphasize the important role that the “target fear level” feature plays in predicting the next level of in-game exposure.

Our results are comparable to those obtained by Liu et al. [12], who used a dynamic difficulty adjustment of game levels based on simple “if” clauses and obtained a classification accuracy of 78%. Having as features both physiological data, Chanel et al. [11] reached a classification accuracy of 63% for the detection of 3 emotional classes in an experiment where 20 participants played a Tetris game with 3 levels of difficulty. Without feature selection, the best classifiers obtained an accuracy of 55% for peripheral features and 48% for EEG features. Feature selection increased the accuracy to 59%, respectively, 56%. Our results are also comparable to those obtained by Lisetti et al. [113], who achieved a classification accuracy of 84% when distinguishing 6 emotional states elicited by movie clips. However, our modality of providing stimuli is more realistic and immersive, as we used for training and testing the classifiers both in vivo and 3D VR stimuli. 

## 9. Conclusions

The purpose of our research was to develop a VR game with ML-based decision support in order to adapt the levels of exposure to the patients’ physiological characteristics. To determine the best ML techniques for acrophobia therapy, several classifiers have been trained: Support Vector Machine, Random Forest, k-Nearest Neighbors, Linear Discriminant Analysis and 4 deep neural network models. We proposed two classifiers: one classifier that estimates the current fear level, based on the user’s physiological recordings and one that predicts the next exposure scenario, i.e., the game level to be played next. We used 3 scales of measuring fear level, with 2, 4 and 11 possible responses (2-choice, 4-choice and 11-choice scale). The validation accuracy is defined as the measure of similarity between the fear level estimated by the first classifier and the Subjective Unit of Distress reported by the user during gameplay. For the 2-choice scale, the highest accuracy has been obtained by DNN_Model_4 (79.12%) for the player-independent modality and SVM (89.5%) for the player-dependent modality. In the case of the 4-choice scale, the highest accuracies were obtained using kNN (52.75%) and SVM (42.5%), respectively. The cross-validation scores are very high for both classifiers, with the best accuracies obtained by the kNN and RF techniques. The most important features for fear level classification were GSR, HR and the values of the EEG in the beta range. For next game level prediction, the “target fear level”, a parameter computed by taking into account the estimated fear level, played a dominant role in classification. 

A future study would be to implement a VR-based game for treating other types of phobias. Moreover, we will extend the experiments and involve more subjects, while their physiological responses will be collected and used for training and testing the classifiers. Another direction we will pursue is to perform real-world tests with the 8 acrophobic patients who participated in the current study, expose them to in vivo scenarios and evaluate whether their anxiety levels dropped.

## Figures and Tables

**Figure 1 sensors-20-00496-f001:**
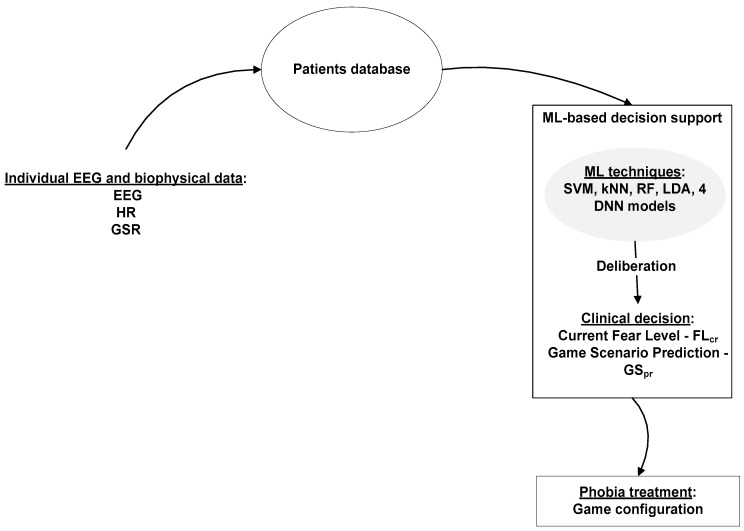
ML-based decision support for phobia treatment.

**Figure 2 sensors-20-00496-f002:**
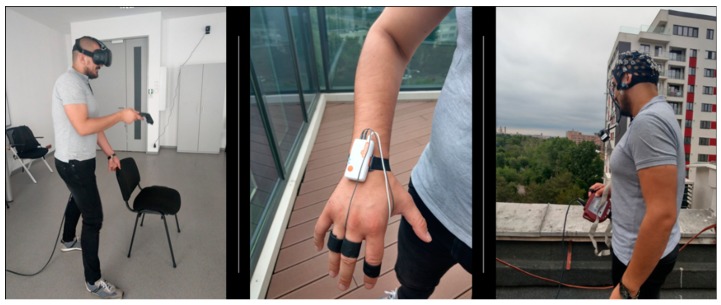
User during in vivo and virtual exposure with physiological signals monitoring.

**Figure 3 sensors-20-00496-f003:**
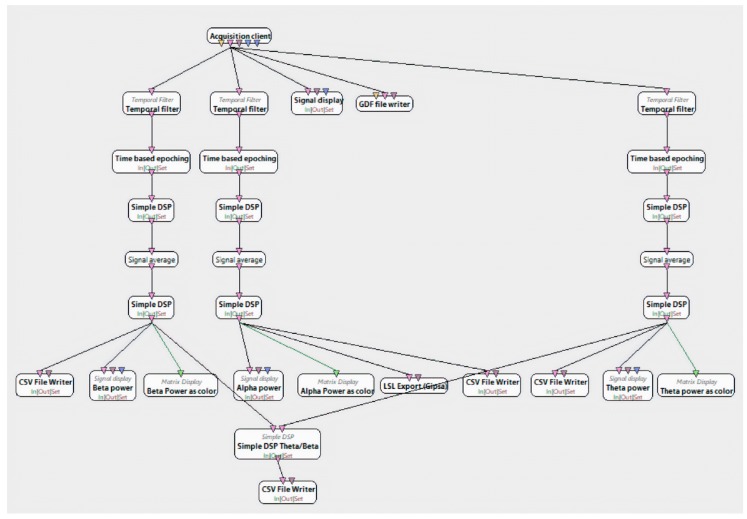
EEG signal recording and decomposition in frequency bands.

**Figure 4 sensors-20-00496-f004:**
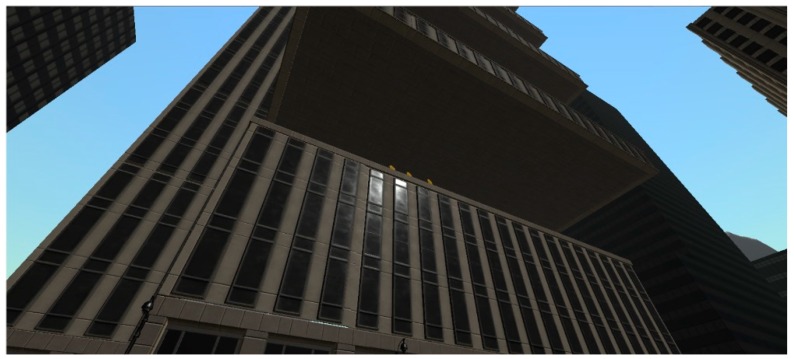
The virtual environment, view of the building from the ground floor.

**Figure 5 sensors-20-00496-f005:**
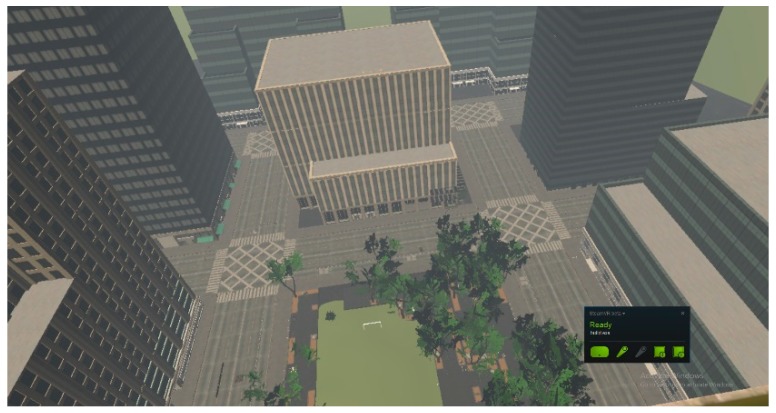
The view from the building’s rooftop.

**Table 1 sensors-20-00496-t001:** Performance in phobia level classification using ML.

	Classifiers	Goal	Signals	Number of Subjects	Performance or Significant Results
[97] 2018	CNN with VGG-16	Detect acrophobia level	EEG	60 subjects	average accuracy 88.77%
[98] 2019	SVM with RBF kernel	Predict anxiety level (public speaking fear)	GSR, BVP, skin temperature	30 persons	BVP accuracy window size 18 s 74.1% GSR accuracy window size 23 s 76.6% Skin temperature accuracy window size 18 s 75.1% Signal fusion (early) window size 20 s 86.3% Signal fusion (late) window size 20 s 83.2%

**Table 2 sensors-20-00496-t002:** Fear level classification scales.

11-Choice-Scale	4-Choice-Scale	2-Choice-Scale
0	0 (relaxation)	0 (relaxation)
1	1 (low fear)	
2		
3		
4	2 (medium fear)	1 (fear)
5		
6		
7		
8	3 (high fear)	
9		
10		

**Table 3 sensors-20-00496-t003:** Properties of the Deep Neural Network models.

DNN Models	Activation Function	Activation Function in the Output Layer	Loss Function	Optimization Algorithm	Epochs and Batch Size
DNN_Model_1 3 hidden layers, with 150 neurons on each hidden	Rectified Linear Unit (RELU)			Adam gradient descent	
layer DNN_Model_2 3 hidden layers, with 300 neurons on each hidden		2-choice scale Sigmoid activation function	2-choice scale Binary crossentropy		1000 epochs for training
layer DNN_Model_3 6 hidden layers, with 150 neurons on each hidden		4-choice scale Softmax activation function	4-choice scale Categorical crossentropy and one-hot encoding		Batch size of 20
layer 6 hidden layers, with 300 neurons on each hidden layer					

**Table 4 sensors-20-00496-t004:** Maximum cross-validation accuracy and test (validation) accuracy (in %) for the player-independent modality, without SFS feature selection.

**Classifier Type**	**C1**				
	**2-Choice Scale**		**4-Choice Scale**		**11-Choice Scale**
	**Cross-Validation**	**Test**	**Cross-Validation**	**Test**	**Cross-Validation**
SVM	80.5	64.75	60.5	46	59.5
kNN	99.5	43.75	99	52.75	98.25
RF	99.25	66.5	99	39.25	99
LDA	79.5	64.75	57.5	37.75	49.25
DNN_Model_1	95	58.3	87.825	45.425	79.4
DNN_Model_2	95.77	58.15	90.525	20.8	84.95
DNN_Model_3	94.75	58.3	86.55	37.7	74.025
DNN_Model_4	94.7	79.12	88.275	37.1	80.85
	**C2**				
	**2-Choice Scale**		**4-Choice Scale**		**11-Choice Scale**
	**Cross-Validation**	**Test**	**Cross-Validation**	**Test**	**Cross-Validation**
SVM	64.25	-	69	-	71
kNN	22.75	-	22.75	-	22.75
RF	99.75	-	100	-	100
LDA	24.5	-	25.75	-	29.5
DNN_Model_1	98.325	-	98.6	-	98.475
DNN_Model_2	98.5	-	98.725	-	98.3
DNN_Model_3	97.675	-	97.825	-	98.325
DNN_Model_4	97.8	-	98.15	-	97.575

**Table 5 sensors-20-00496-t005:** Maximum cross-validation accuracy and test (validation) accuracy (in %) for the player-independent modality, with SFS feature selection.

**Classifier Type**	**C1**				
	**2-Choice Scale**		**4-Choice Scale**		**11-Choice Scale**
	**Cross-Validation**	**Test**	**Cross-Validation**	**Test**	**Cross-Validation**
kNN	54	49.9175	32.25	30.24	25
RF	54.5	60.4175	33.25	38.5725	29.75
LDA	65.75	64.585	35.25	33.5725	25.25
	**C2**				
	**2-Choice Scale**		**4-Choice Scale**		**11-Choice Scale**
	**Cross-Validation**	**Test**	**Cross-Validation**	**Test**	**Cross-Validation**
kNN	32.75	-	36	-	41.75
RF	35.5	-	40.5	-	41.75
LDA	37.25	-	42.75	-	44.5

**Table 6 sensors-20-00496-t006:** Maximum cross-validation accuracy and test (validation) accuracy (in %) for the player-dependent modality, without SFS feature selection.

**Classifier Type**	**C1**				
	**2-Choice Scale**		**4-Choice Scale**		**11-Choice Scale**
	**Cross-Validation**	**Test**	**Cross-Validation**	**Test**	**Cross-Validation**
SVM	88	89.5	74.75	42.5	77.75
kNN	99.5	77	99	29.25	98.25
RF	99.75	77	99.25	21	99
LDA	87	60.5	71.25	21.75	64
DNN_Model_1	95.03	72.9	87.945	41.8975	79.485
DNN_Model_2	95.51	68.735	90.4975	24.9925	85.095
DNN_Model_3	94.4375	62.45	86.325	34.15	74.275
DNN_Model_4	94.575	54.125	88.28	38.325	80.45
	**C2**				
	**2-Choice Scale**		**4-Choice Scale**		**11-Choice Scale**
	**Cross-Validation**	**Test**	**Cross-Validation**	**Test**	**Cross-Validation**
SVM	82.75	-	86.5	-	86.5
kNN	23.75	-	23.75	-	23.75
RF	99.75	-	99.75	-	100
LDA	23	-	20.5	-	27.5
DNN_Model_1	98.4	-	98.675	-	98.75
DNN_Model_2	98.725	-	98.5	-	98.65
DNN_Model_3	97.45	-	97.825	-	98.5
DNN_Model_4	97.375	-	97.775	-	98.175

**Table 7 sensors-20-00496-t007:** Maximum cross-validation accuracy and test (validation) accuracy (in %) for the player-dependent modality, with SFS feature selection.

**Classifier Type**	**C1**				
	**2-Choice Scale**		**4-Choice Scale**		**11-Choice Scale**
	**Cross-Validation**	**Test**	**Cross-Validation**	**Test**	**Cross-Validation**
kNN	76.75	72.9175	52.25	16.665	42
RF	77	68.75	49.75	28.5725	45.75
LDA	81	85.4175	54.5	17.5	40.5
	**C2**				
	**2-Choice Scale**		**4-Choice Scale**		**11-Choice Scale**
	**Cross-Validation**	**Test**	**Cross-Validation**	**Test**	**Cross-Validation**
kNN	50.25	-	52.25	-	53.25
RF	50.5	-	53.5	-	56.5
LDA	52	-	56	-	56.75

**Table 8 sensors-20-00496-t008:** Feature (F) and feature importance (FI) for the player-independent modality.

C1						C2					
2-Choice Scale		4-Choice Scale		11-Choice Scale		2-Choice Scale		4-Choice Scale		11-Choice Scale	
F	FI	F	FI	F	FI	F	FI	F	FI	F	FI
GSR	0.41	GSR	0.45	GSR	0.49	GSR	0.44	FL_t_	0.69	FL_t_	0.87
HR	0.28	HR	0.28	HR	0.24	FL_t_	0.37	GSR	0.41	GSR	0.39
B_C3	0.15	B_FC6	0.15	B_FC6	0.14	HR	0.23	HR	0.20	HR	0.18
B_P3	0.13	B_C3	0.13	B_FC5	0.12	B_FC6	0.14	A_FC6	0.12	B_FC6	0.13
B_FC2	0.13	B_FC2	0.12	B_C3	0.12	A_FC6	0.13	B_FC6	0.12	A_FC6	0.11
B_FC6	0.13	B_FP1	0.12	B_FC2	0.12	B_FC5	0.10	B_P3	0.10	B_P3	0.09
B_FP2	0.12	B_P3	0.12	B_P3	0.11	T_FC6	0.10	B_T8	0.09	B_FC2	0.09
A_FC6	0.12	T_FC6	0.12	T_FC6	0.11	B_P3	0.09	B_FC2	0.09	T_FC6	0.08
B_C4	0.10	B_O1	0.11	B_FP1	0.10	B_T8	0.09	B_C3	0.09	B_T8	0.08
B_FC5	0.10	B_FC5	0.11	A_FC6	0.10	B_O1	0.09	T_FC6	0.08	B_FC5	0.07
B_FP1	0.09	B_T8	0.09	B_T8	0.10	B_C3	0.09	B_O2	0.08	B_O2	0.07
T_FC6	0.08	B_P2	0.09	B_O1	0.08	B_FC2	0.09	B_FC5	0.08	B_FP1	0.07
A_FP1	0.08	B_FC1	0.08	A_FP1	0.08	B_O2	0.09	B_FP1	0.07	B_C3	0.07
A_FP2	0.08	A_FP1	0.08	B_P2	0.08	B_P2	0.08	A_FP1	0.07	B_P2	0.06
B_T8	0.08	A_FC6	0.08	T_FP1	0.08	B_FP1	0.08	A_O1	0.06	B_O1	0.06

**Table 9 sensors-20-00496-t009:** Feature (F) and feature importance (FI) for the player-dependent modality.

C1						C2					
2-Choice Scale		4-Choice Scale		11-Choice Scale		2-Choice Scale		4-Choice Scale		11-Choice Scale	
F	FI	F	FI	F	FI	F	FI	F	FI	F	FI
GSR	0.40	GSR	0.46	GSR	0.48	GSR	0.54	FL_t_	0.66	FL_t_	0.79
HR	0.25	HR	0.32	HR	0.27	FL_t_	0.32	GSR	0.47	GSR	0.42
B_FC2	0.22	B_FC6	0.17	B_FP1	0.14	HR	0.24	HR	0.20	HR	0.18
B_C4	0.15	B_FC2	0.16	A_FC6	0.14	A_FC6	0.15	B_FC6	0.14	T_FC6	0.12
B_FC6	0.14	B_P2	0.12	B_FC2	0.14	B_FC6	0.14	A_FC6	0.11	B_FC6	0.12
A_FP1	0.14	B_FP1	0.12	B_FC6	0.13	B_FP1	0.12	B_FC2	0.10	A_FC6	0.12
B_P2	0.13	T_FC6	0.11	T_FC6	0.12	T_FC6	0.10	T_FC6	0.09	B_P3	0.11
A_FC6	0.12	B_O1	0.10	B_O1	0.12	B_FC2	0.10	B_FC5	0.08	B_FC2	0.11
B_FP1	0.10	A_FC6	0.10	A_FP1	0.11	B_O2	0.09	B_O2	0.08	B_FP1	0.08
B_O2	0.10	A_FP1	0.10	B_FC5	0.11	B_P1	0.09	B_C4	0.08	A_FC1	0.08
T_P2	0.08	B_P3	0.10	B_P2	0.10	B_O1	0.08	B_FP1	0.07	T_FP1	0.07
T_FC6	0.08	B_C4	0.09	B_P3	0.10	A_O1	0.08	A_P4	0.07	A_O1	0.07
B_O1	0.08	B_FC5	0.09	B_C3	0.09	B_P2	0.08	A_FP1	0.07	B_T8	0.07
B_C3	0.08	A_P2	0.08	B_T8	0.09	T_P3	0.07	B_P2	0.07	B_C4	0.07
B_P3	0.08	B_C3	0.08	B_C4	0.08	A_P2	0.07	B_C3	0.07	B_O2	0.07

**Table 10 sensors-20-00496-t010:** Highest cross-validation and test accuracies.

Method	C1				
	2-Choice Scale		4-Choice Scale		11-Choice Scale
	Cross-Validation	Test	Cross-Validation	Test	Cross-Validation
Player-independent	kNN 99.5%	DNN_Model_4 79.12%	kNN 99%	kNN 52.75%	kNN 98.25%
	RF 99.25%		RF 99%		RF 99%
Player-dependent	kNN 99.5%	SVM 89.5%	kNN 99%	SVM 42.5%	kNN 98.25%
	RF 99.75%		RF 99.25%		RF 99%
	C2				
	2-Choice Scale		4-Choice Scale		11-Choice Scale
	Cross-Validation	Test	Cross-Validation	Test	Cross-Validation
Player-independent	RF 99.75%	-	RF 100%	-	RF 100%
Player-dependent	RF 99.75%	-	RF 99.75%	-	RF 100%

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
