# Peer review of "An Investigation of Various Machine and Deep Learning Techniques Applied in Automatic Fear Level Detection and Acrophobia Virtual Therapy"

_sensors, 2020, doi:10.3390/s20020496_

Round 1
Reviewer 1 Report
The authors presented a very interesting work of using ML techniques on biophysical and EEG data to facilitate the decision-making process of a virtual reality exposure therapy. The idea is quite innovative, however, there are still a few things in the manuscript that need to be addressed before being accepted for publication.
First, the profiles and demographics of the patients are not well presented. There is a lack of detailed info such as the patients’ ages, genders, whether receiving other forms of treatment or not (including long-term medications / substance usage), etc. Without the mentioned information, the validity of the experiment could be challenged and a reproduction of similar results on other patients for future research could be difficult.
Second, the data collection process by using the various sensors is not clearly demonstrated. How are the data from different sources / sensors aligned on a single timeline? And how is the timeline aligned with the presentations of the stimuli in virtual reality? For the second question, it is a little confusing since the biophysical data were sampled at an interval of 65ms (appx. 15.38Hz) while the HTC VIVE headset used in the experiment refreshes at 90Hz. Moreover, the raw data tend to be noisy. Any method (e.g., filtering) was used for preparing and processing the raw data? The authors mentioned data processing in line 501 - 502 on page 14, but there’s a lack of clear explanation on the details, so the authors are also suggested to use formulas to present such processing.
Third, the detailed design of the virtual reality game needs to be presented. Were the patients able to move freely in the virtual reality scenes? Besides the graphical stimuli (presented in Figure 3 and 4), any other forms of stimuli were accompanied (e.g., the sound of wind and traffics, etc.).
Author Response
Dear reviewer,
Please find attached the cover letter that contains detailed point-by-point responses to your comments.
All the best,
The authors

Reviewer 2 Report
Authors tackled the problem of semi-automatic VR phobia (acrophobia) treatment. They developed a VR game controlled with a controllers analysing biophysical signals (fear determinants). Though the problem is valid and interesting, provided manuscript reveals severe methodology flaw and inconsistencies.
Authors’ definition of affective computing notion (lines 62-63) is very narrow and in my opinion evidently goes beyond systems which “adapt to the users’ affective cues”. Further analysis of the state-of-the-art and subsequent sections becomes difficult to follow.
In the introduction authors introduce elements of the methodology (lines 74-82) which become rather a standard of training within-subject and between-subjects classification, thus their discussion in the first section of the manuscript is intricate.
Statement that “the most relevant features for fear level classification were GSR, HR …” becomes problematic and ambiguous, because the whole GSR signal and Heart Rate Value signal become a source of almost countless specific features. In case of “beta frequency range” how the “value of the signal” was explicitly defined?
Authors recalled a notion of a “target fear level” (line 91), dependant probably on some model conditioning “fear levels” and “subsequent exposure scenarios” (line 70) but it was not cleared in the manuscript. Moreover authors did not motivate necessity of deep learning exploitation in a “fear modelling” task. Additionally C2 classifier (lines 366-367) seems to be a trivial task if an appropriate fear level was determined with C1 classifier and adequate medical fear-alleviating treatment is known in advance.
In the state-of-the-art section authors provide shallow review of recent VR systems without adequate critical analysis of the reference systems in the context of sensors modality, classification methods which become a presumed contribution of the manuscript. Pure functional, content-related VR systems analysis does not provide adequately motivation for authors’ research. Neither platforms, nor academic research projects and mobile/desktop game applications description specify aspects of user-centred scenarios adaptability, selected communication modalities deficits. It mainly concentrates on a well known facts that VR treatment can alleviate different phobias. In this context several important questions arise:
- how affective states are monitored in reference applications?
- what types of affective states are monitored in reference applications?
- why machine learning methods are required for fear level related scenarios assignment?
- why other therapeutic solutions were unreliable in phobias alleviation as authors declare providing reliable solution (line 163)?
Analysis of emotion models and biophysical data (section 3) should be summarised and supplemented with some selection of features adequate for fear state classification. It should appear as a conclusion of the reference researches analysis.
Analysis of ML methods for emotion recognition is quite general and concentrates on output metrics rather then effectiveness of features selection or evaluation of ML methods in the context of their advantages/disadvantages for emotion recognition. Moreover, problem of VR-based emotion recognition should be analysed in a context of VR head-sets and dedicated controllers constraints which evidently may interrupt a signal acquisition process.
On the other hand, I do believe that aspect of multi-level emotional states detection (i.e. 4-choice scale, 11-choice scale) is very important and rather scarce in literature of the subject, but provided state-of-the-art does not reflect this specific issue.
Method description is a bit confusing.
- why in the Figure 1 “clinical decision” does not become an input of “Phobia treatment” module?
- why the relation of current and target fear levels is looped/reversed, i.e. (in 2-choice scale) current =1 then target =0 and (in 4-choice scenario) current=3 then target=current-1?
- some explanations become inconsistent, as on one hand authors claim that (lines 386-387) “The next exposure scenario has been predicted in real-time by C2, based on the EEG, biophysical data and the target fear level”, whereas (line 362) “C1 estimates the level of fear the patient currently experiences” and provides an input for C2 which (line 367) “estimates the phobia treatment.”. Probably detailed scheme of the method should be provided. If some other signal, beyond current fear level is interpreted by C2, the motivation behind it should be supplemented.
- do the experiments exposure heights (1st, 4th, 6th floor, rooftop) depend on any medical treatment procedures?
- how precisely input features of the biophysical data were defined, as normalised power of EEG channels, in my opinion, is not precise and becomes ambiguous?
- description of classifiers, verified in the experiments, is quite general as they comprise a lot of crucial, inherent parameters which were not motivated;
- from the initial part of the manuscript, it is not clear that C2 classifier receives as an input EEG, GSR, HR and SUD values as the C1 classifier does as well - according to authors initial declaration, biophysical signals were used for training C1 classifier. If so, the question about the motivation for such a doubled architecture of classifiers arises. Why the biophysical signals were analysed twice?
- how the classification quality metrics were defined? How the reference, describing a ground truth treatment scheme was designed? How the test accuracy (line 511) was defined?
- what type of “features selection” (line 492) authors mean if the in lines 428-429 there was no information about possible alternative features selection approaches?
- why the C2 accuracy was not evaluated as it was presumably one of the main contributions of the manuscript?
As a consequence of all previously mentioned imprecisions it is difficult to determine whether obtained results are reliable. High accuracy values were obtained in an independent (made by authors) game scenario thus it can not be directly compared with referenced researches of Liu, Chanel, Lisetti (lines 602-611).
Final conclusions, suggesting “determination of the best ML technique for acrophobia therapy” (lines 614-615), in my opinion, was not supported by provided results.
Besides quite numerous and extensive bibliography, most of provided positions are of minor importance (loosely connected http addresses) or outdated references (older then 3 years).
From the language point of view the manuscript is written rather well, though still some minor imprecisions can be found, i.e. how the “performance of classifiers” (lines 489-490) was defined?
Summing up, chaotic structure of the manuscript, severe methodology flaws, weakness of the contribution and all pointed doubts incline my opinion towards rejection of the manuscript in a current form.
Author Response

(The authors gave the same response as above.)
